# Neural Machine Translation with Soft Prototype

**Yiren Wang**[1],[*] **Yingce Xia**[2],[†] **Fei Tian**[3], **Fei Gao**[4], **Tao Qin**[2], **ChengXiang Zhai**[1], **Tie-Yan Liu**[2]
[1]University of Illinois at Urbana-Champaign, [2]Microsoft Research,
[3]Facebook, [4]Institute of Computing Technology, Chinese Academy of Sciences
[1]{yiren, czhai}@illinois.edu  [2]{yingce.xia, taoqin, tie-yan.liu}@microsoft.com
[3]feitia@fb.com [4]gaofei17n@ict.ac.cn

## Abstract

Neural machine translation models usually use the encoder-decoder framework and generate translation from left to right (or right to left) without fully utilizing the target-side global information. A few recent approaches seek to exploit the global information through two-pass decoding, yet have limitations in translation quality and model efficiency. In this work, we propose a new framework that introduces a soft prototype into the encoder-decoder architecture, which allows the decoder to have indirect access to both past and future information, such that each target word can be generated based on the better global understanding. We further provide an efficient and effective method to generate the prototype. Empirical studies on various neural machine translation tasks show that our approach brings substantial improvement in generation quality over the baseline model, with little extra cost in storage and inference time, demonstrating the effectiveness of our proposed framework. Specially, we achieve state-of-the-art results on WMT2014, 2015 and 2017 English→German translation.

## 1  Introduction

Neural machine translation (briefly, NMT) [1, 17, 16] has attracted much research attention recently. The major approach for neural machine translation (also, sequence generation) is to generate a target sequence through a one-pass decoding process, either from left to right [17, 1] or from right to left [15]. To translate a sentence $x$, the model usually conditions on the source sentence $x$ and previously generated words $y_{<t}$ to generate the $t$-th target word $y_t$.

One limitation of such a one-pass process is that the generation of word $y_t$ only uses partial information of the previously generated incomplete sentence $y_{<t}$, rather than considering the global information carried by a complete possible target candidate $y' = (y'_1, y'_2, ..., y'_l)$. Global information of the target domain is intuitively beneficial for sequence generation, since each word in the target sentence has to be consistent with its surrounding words, i.e. both words before and after it. Different approaches have been proposed to leverage the global information to overcome such limitation, including introducing additional networks that generate an intermediate sequence $y'$ from left to right [18, 12, 20] or from right to left [21], and retrieving an intermediate sequence $y'$ from an existing corpus [5, 4, 11]. These approaches, though proposed for different tasks under different settings, share the same intrinsic idea to introduce a prototype sequence to the standard encoder-decoder framework. For example, the intermediate sequence $y'$ in [4, 5] serves as a prototype. When generating each token $y_t$, the prototype allows the decoder to have indirect access to both previous ($y'_{<t}$) and future ($y'_{>t}$) information, and thus can generate all tokens based on a good global understanding.

---

[*]This work is conducted at Microsoft Research Asia.
[†]Corresponding author.

Intuitively, a good prototype should have the following properties: (1) *Good quality*: an ideal prototype should be a high-quality representation with both rich global information and low variance; and (2) *High efficiency*: the prototype should be easy to generate, with little additional cost in time and storage. The previous prototypes, which are obtained either by an additional decoding step [18, 12, 20] or by retrieving from a large corpus [4, 5, 11], are of relatively good quality with complete global information, but very inefficient to generate. Furthermore, because of the high cost of prototype generation, most previous approaches use a single sequence as a *hard* prototype in practice, leading to high variance that might deteriorate the quality of the eventual output.

In this paper, we propose a new framework with *soft* prototype to leverage the target side global information. The soft prototype $R$, which is the key component of the proposed framework, is a global representation calculated by multiple (instead of a single) candidates in the target domain. Both the source sentence $x$ and the soft prototype $R$ are encoded into higher-level contextual representations, which are then decoded to generate a target sentence. We further introduce a simple yet effective solution to efficiently obtain a soft prototype. Specifically, each word in the input sequence is mapped into a distribution over the target vocabulary, and the weighted average of target word embeddings is treated as an "expected" word representation in the prototype $R$. The probabilistic manner of this solution helps to reduce variance and enrich the encoded information, and the word-level mapping in the non-autoregressive manner guarantees efficiency.

Our framework is general and can build on any model architectures for sequence generation, including LSTM [17], CNN [3], Transformer [16], etc. In this work, we integrate the proposed soft prototype into the state-of-the-art Transformer [16] model, and evaluate it on multiple NMT tasks with different data settings, including the supervised, semi-supervised and unsupervised NMT. Experiments show that our method significantly improves the translation quality over the strong Transformer baselines. Furthermore, integrating our soft prototype with sophisticated translation systems results in the state-of-the-art performances for WMT2014, 2015 and 2017 English-to-German translation tasks.

## 2 Background

Given two domains $\mathcal{X}$ and $\mathcal{Y}$ associated with vocabularies $\mathcal{V}_x$ and $\mathcal{V}_y$ respectively, we denote $x = (x_1, x_2, \cdots, x_{l_x}) \in \mathcal{X}$ as the source sentence with $l_x$ tokens $x_i \in \mathcal{V}_x$, and $y = (y_1, y_2, \cdots, y_{l_y}) \in \mathcal{Y}$ as the target sentence with $l_y$ tokens $y_j \in \mathcal{V}_y$, $i \in [l_x]$, $j \in [l_y]$. A sequence generation model aims to learn a mapping $f : \mathcal{X} \mapsto \mathcal{Y}$. In the context of neural machine translation, $\mathcal{X}$ and $\mathcal{Y}$ are collections of sentences in two different languages.

Most NMT systems adopt the encoder-decoder framework [15], where the input sequence $x \in \mathcal{X}$ is firstly processed by the encoder Enc to get higher-level context representations, which are then fed into the decoder Dec to generate the output $y \in \mathcal{Y}$, i.e., $y = \text{Dec}(\text{Enc}(x))$. Enc and Dec can be specialized using different neural architectures including GRU [1], LSTM [17], CNN [3], and Transformer [16], among which the recent self-attention based Transformer is the state-of-the-art architecture for NMT.

Previous approaches that leverage the target side global information for better generation quality are mostly based on the following formulation:

$$P(y|x) = \sum_{y' \in \mathcal{Y}} P(y|y', x; f_e) P(y'|x; g), \tag{1}$$

where $g$ is a generator mapping the input $x$ to a coarse $y'$, and then the model $f_e$ further deliberates/refines $y'$ to $y$. $f_e$ usually consists of three parts: two encoding modules used to extract hidden representations of $x$ and $y'$, and one decoding module that maps the hidden representations to the eventual output. The differences of previous works lie in how $g$ is designed: as a sequence generator that decodes in the left-to-right [18, 12, 20] or right-to-left [21] manner; or as a retriever that samples sequences from a training corpus similar to $x$ [5, 4, 11].

Directly generating sequences following Eqn.(1) is infeasible given that $\mathcal{Y}$ is exponentially large (since it represents all possible sequence in the target domain). Therefore, previous works use a Monte Carlo based estimation instead, and usually estimate with a single sample $y'$ in the actual implementation. That is, sample a $y'$ from distribution $P(\cdot|, x; g)$ and then approximate $P(y|x)$ by $P(y|x) \approx P(y|y', x; f_e)$.

# 3   Our Approach

In this section, we first introduce the general framework with soft prototype (Section 3.1). We then introduce a specific solution to generate an effective soft prototype efficiently (Section 3.2), and show how the proposed prototype is integrated to the Transformer [16] model (Section 3.3).

## 3.1   Framework

The general framework with prototype for sequence to sequence learning can be formulated as:

$$y = \texttt{Dec}(\texttt{Enc}(x), \texttt{Net}(R)). \tag{2}$$

where $\texttt{Enc}$ and $\texttt{Dec}$ are the encoder and decoder networks same as those in traditional sequence to sequence framework, $\texttt{Net}$ is the additional network that encodes the prototype $R$ to higher level contextual representation. The prototype $R$ is a matrix $(r_1, r_2, ..., r_l)^\top$ with size $l \times d$, where each row $r_i$ is a vector calculated based on one or multiple $y' \in \mathcal{Y}$, $l$ is the length of the prototype and $d$ is the dimension of the hidden representation.

Define $E_y$ as the embedding of target domain with size $|\mathcal{V}_y| \times d$, where $d$ is defined as the dimension of embedding (same as the prototype). Given any $y' \in \mathcal{Y}$, let $\mathbf{1}(y')$ denote the corresponding 2D representation with size $l_{y'} \times |\mathcal{V}_y|$, where $l_{y'}$ is the length of $y'$. Each row $t$ is a one-hot vector of the word $y_t$, $t \in [l_{y'}]$. The soft prototype is calculated as:

$$R = \sum_{y' \in \mathcal{Y}} \mathbf{1}(y') E_y P(y'|x; g) \stackrel{\text{def}}{=} G_y(x) E_y \tag{3}$$

where $g$ is the generator[3]. We define $G_y(x) = \sum_{y' \in \mathcal{Y}} y' P(y'|x; g)$, where $G_{y,ij}$ intuitively is the probability that the $i$-th representation in the prototype is the embedding of the $j$-th word in $\mathcal{V}_y$.

Our framework is a biased approximation of Eqn.(1). As explained in Section 2, Eqn.(1) is intractable to calculate due to the following two reasons: i) $|\mathcal{Y}|$ is exponentially large since $\mathcal{Y}$ represents all possible sentences in the target domain; ii) $f_e$ is usually specialized as deep neural networks, which makes it even more costly to compute $P(\cdot|y', x; f_e)$. Our framework alleviates such issues with an efficient approximation, that is, first aggregating the potential candidates through the soft prototype $R$ calculated by Eqn.(3) and then generating $y$ with a single model by Eqn.(2). We only need to calculate the probability using model $f_e$ once, instead of $|\mathcal{Y}|$ times as shown in Eqn.(1).

Previous approaches usually adopt an approximation of Eqn.(1) by generating/retrieving only one $y'$ in their practical implementations to reduce the high computational cost but introduce high variance. In such cases, their methods fall into our framework as different solutions to generate the prototype with different $G_y(x)$. Specifically, they all construct the probabilistic matrix $G_y(x)$ with each row $g_i$, $\forall i \in \{1, 2, ..., l\}$, as a one-hot vector, which is constructed either by autoregressively decoded sentences [18, 21] or retrieved sequences [5, 11].

Reformulating the previous approaches under our framework in this way offers us a better understanding of quality-efficiency trade-off in the solution design: (A) $G_y$ as a one-hot representation is simple to construct, yet leads to high variance and information loss that potentially limits the quality of the final generated output. (B) $G_y$, which is generated by autoregressive decoding or retrieval, provides a context-aware representation, yet leads to the common issue of low efficiency. The cost of generation based approaches [18, 21, 12] comes from the auto-regressive decoding process, and the complexity of retrieval based approaches [5, 4] comes from searching in exponentially large space of sequences[4].

## 3.2   Efficient Generation of Soft Prototype

A good prototype should have good quality with rich global information and low variance, plus high efficiency. Eqn.(3) satisfies the first one, but in general, still suffers from low efficiency considering $|\mathcal{Y}|$ is exponentially large and the tokens of $y'$ are sequentially generated one by one. This motivates us to use a non-autoregressive way.

To achieve that, we use a probability generator $g$, that projects any $v_x \in \mathcal{V}_x$ to a $(|\mathcal{V}_y|-1)$-dimensional simplex. That is, for any $v_x \in \mathcal{V}_x$, $g(v_x) = (p_1, p_2, \cdots, p_{|\mathcal{V}_y|})$, where $p_j \geq 0$ and $\sum_{j=1}^{|\mathcal{V}_y|} p_j = 1$. In this case, given any $x$, $G_y$ is a $l_x \times |\mathcal{V}_y|$ matrix, where the $l_x$ is the number of words in $x$. The $i$-th row of $G_y$ is $g(x_i)$, for any $i \in [l_x]$. Then the prototype $R$ can be calculated by Eqn.(3) accurately.

The non-autoregressive method has several advantages over the previous method: A) Better efficiency: There is no need to sum over an exponentially large space $|\mathcal{Y}|$ with the auto-regressive property removed; and B) Richer information: Previous approaches only refines on a hard prototype represented by a single $y'$, while our soft prototype consists multiple candidate translations, which contributes to richer information and lower variance.

Note that when $\mathcal{V}_y$ is large, saving and estimating $g$ becomes more expensive. We adopt an approximation strategy here to alleviate this issue with a sparse probability generator $g^\kappa$, where $\kappa \in \mathbb{N}$. For each input, $g^\kappa$ will only output the $\kappa$-largest probabilities, zeroing out the remaining elements. The outputs are then normalized to sum $1$ as the final probability to generate the soft prototype.

### 3.3 Adaptation with Transformer

We show how the proposed framework with soft prototype is adapted to the Transformer architecture. The modified model is illustrated in Figure 1. We omit the details of Transformer and give a high-level introduction here. The source codes are included in the supplementary documents and details can be found at `transformer_softproto.py`.

A) The soft prototype $R$ is generated with the probabilistic generator $g$ (or $g^\kappa$) following Eqn.(3). The prototype is then encoded into higher-level global representations by $\tilde{H} = \texttt{Net}(R)$, while the input sentence $x$ is encoded by $H = \texttt{Enc}(x)$. Both $\texttt{Enc}$ and $\texttt{Net}$ are specialized as $L$-layer Transformer encoders (module $\texttt{Net}$ is in lower right part of the figure).

B) The two context representations $H$ and $\tilde{H}$ are then used as context vectors for the encoder-decoder attention in the Transformer decoder. Specifically, let $s_t^l$ denote the output of the $l$-th decoder layer at step $t(>0)$, and $s_j^0$ denote the embedding vector of the $j$-th generated word in the target domain. Let $F_S(q, K, V)$,

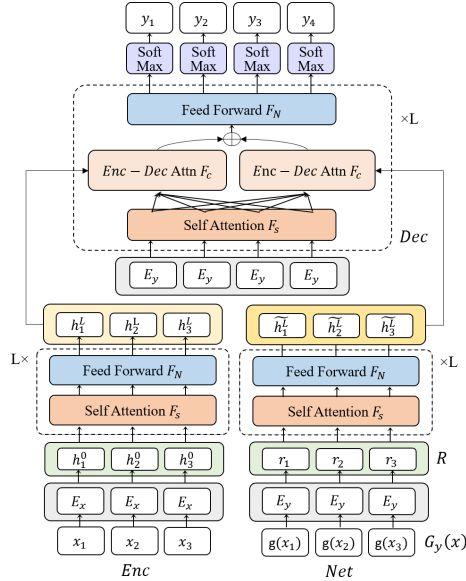

Figure 1: Illustration of the modified Transformer with soft prototype.

$F_C(q, K, V)$ and $F_N(h)$ denote the self attention layer, the encoder-decoder attention layer, and the feed-forward layer respectively. A self-attention layer requires a query $q$, key $K$ and value $V$ as inputs. (Detailed formulation of the aforementioned three layers is left in Appendix A). Each $s_t^l$ is obtained as follows:

$$
\begin{aligned}
\tilde{s}_t^l &= F_S(s_t^{l-1}, S_t^{l-1}, S_t^{l-1}); \\
\hat{s}_t^l &= F_C(\tilde{s}_t^l, H, H) + F_C(\tilde{s}_t^l, \tilde{H}, \tilde{H}); \\
s_t^l &= F_N(\hat{s}_t^l),
\end{aligned}
\tag{4}
$$

where $S_t^{l-1} = \{s_1^{l-1}, s_2^{l-1}, \cdots, s_t^{l-1}, \cdots\}$. An affine transformation is later applied on $s_t^L$ to generate word $y_t$. For the sake of storage and training efficiency, we set $\texttt{Enc} = \texttt{Net}$ in our experiments. A detailed study on parameter reuse is presented in Appendix B.

#### Discussion

1) *Storage Cost*: With parameter reuse (i.e., set $\texttt{Enc} = \texttt{Net}$), the total number of network parameters is exactly the same as the standard transformer. The only additional storage cost comes from the probabilistic generator $g^\kappa$, which uses additional $2\kappa|\mathcal{V}_y|$ parameters to store the prototype distribution

in the form: {(non-zero word id: probability)}, which is almost negligible compared to the large NMT model.

2) *Inference Efficiency*: In the standard Transformer, a large portion of inference time is spent on the decoding process, as the encoding stage has less operations and allows parallelism. Therefore, the new approach to generate the soft prototype, which introduces more complexity into the encoder side, will not greatly hamper the inference efficiency like the previous generation base methods. Compared with the standard model, our method introduces $34\%$ additional inference time while DelibNet brings more than $80\%$. We present detailed analysis on storage cost and inference efficiency in Section 4.1.2.

3) *Token-level translation*: Token-level mapping is employed here for the maximum efficiency. While there are potential issues such as translating "middle" BPE tokens, their impact is relatively small given that: (a) the vocabulary and training corpus is dominant by the standard words rather than "middle" tokens. For example, over $65\%$ vocabulary in WMT14 En→De are standard words and they make up for over the $88\%$ of total word frequency in the training corpus; (b) The soft prototype $R$ is fed to `Net` and encoded into higher-level contextual representations, which can intuitively provide rich global information that helps the decoder decision making.

## 4 Experiments

In this section, we present empirical studies for the proposed framework on multiple tasks across different setting, including the supervised, semi-supervised and unsupervised NMT.

### 4.1 Supervised NMT

We first study the effectiveness of the framework under the supervised setting with aligned bilingual data only.

**Datasets** We experiment with two large scale and widely adopted benchmark datasets, the WMT2014 English to German news translation (En→De) and WMT2014 English to French news translation (En→Fr). We use $4.5M$ bilingual sentence pairs as the training data for En→De and $36M$ pairs for En→Fr[5]. We use the concatenation of Newstest2012 and Newstest2013 as the validation set (6003 sentences) and Newstest2014 as the test set (3003 sentences). All words are split into subword units using byte pair encoding (BPE) [14], forming a vocabulary of $32k$ and $45k$ for En→De and En→Fr respectively. The vocabulary is shared among source and target languages.

**Model** We use the `transformer_big` setting following [16], with a 6-layer encoder and 6-layer decoder. The dimensions of word embeddings, hidden states and the filter sizes are 1024, 1024 and 4096 respectively. The dropout is $0.3$ for En→De and $0.1$ for En→Fr. The models are trained on 8 M40 GPUs for 10 days for En→De and 21 days for En→Fr. $\kappa$ is fixed as 3 across all tasks. We use beam size 4 and length penalty 0.6 for inference, and use `multi-bleu`[6] to evaluate the quality of translation.

**Optimization** We use Adam [7] with same learning rate scheduler used in [16] for optimization. The model with our prototype is initialized as follows: (1) A standard NMT model $f_0$ is pre-trained by maximizing $\sum_{(x,y)\in\mathcal{X}\times\mathcal{Y}}\log P(y|x;f_0)$. The stopping criterion is that when the validation BLEU does not improve in $10k$ iterations, output $f_0$. (2) Define $g_{ij} = P(w_j|v_i;f_0)$ where $v_i \in \mathcal{V}_x$ and $w_j \in \mathcal{V}_y$; convert $g$ to the sparse probabilistic probabilisitc generator $g^\kappa$; (3) The modified Transformer with soft prototype is trained with $g^\kappa$ and use $f_0$ to initialize the new model.

### 4.1.1 Results

The results for En→De and En→Fr translations are presented in Table 1. We compare the performances of multiple systems: (1) The Transformer baseline (*Transformer*); (2) model with hard prototype, including DelibNet with stack LSTM (*DelibNet*) reported in [18] and our implemented DelibNet with Transformer (*Transformer + DelibNet*) for fair comparison; and (3) transformer with soft prototype (*Transformer + SoftPrototype*).

Table 1: Results on WMT2014 En→{De, Fr} translations. † denotes the results of our own implementation, which is slightly better than reported values in [16]. "#Params" denotes the number of model parameters for En→De. "Inference time" is measured as the total decoding time on one M40 GPU for En→De newstest2014 (3003 sentences), with a batch size of 128 and beam size 4.

| | En→De | En→Fr | #Params | Inference time |
|---|---|---|---|---|
| Transformer [16] | $28.4 \,/\, 28.68^{\dagger}$ | $41.8 \,/\, 41.85^{\dagger}$ | $200M$ | 116s |
| DelibNet [18] | − | 41.50 | − | − |
| Transformer + DelibNet | 29.11 | 42.58 | $372M$ | 207s |
| **Transformer + SoftPrototype** | **29.46** | **42.99** | $200.2M$ | 156s |

From the results, we observe that: (1) Introducing the target side prototype to leverage the global information can significantly boost the model performance. Different prototypes, either in the hard manner or soft, lead to improvements in translation quality that are statistically significant with $p < 0.01$ in paired bootstrap sampling[7] [8]. (2) The proposed approach for generating the soft prototype achieves better performances with better efficiency, compared with the two-pass decoding DelibNet. We present more detailed study and discussion in section 4.1.2.

### 4.1.2 Analysis

We present multiple studies to thoroughly analyze the proposed framework with soft prototype, including training/inference efficiency and translation quality.

**Inference Efficiency.** We report the number of model parameters and the total inference time for decoding 3003 sentences in Table 1. As we can see: (1) DelibNet with an extra decoder and softmax layer introduce $86\%$ additional parameters to the standard Transformer, and increase the inference time by $80\%$. (2) Transformer+SoftPrototype is the most efficient way in terms of both inference time and storage cost, with only $0.1\%$ additional storage for $g^{\kappa}$. The method doubles the encoding time to encode both source sequence and the prototype. As encoding is far less expensive than decoding during inference in NMT, the increase in inference time is smaller ($34\%$).

**Training Efficiency.** Since we use the pre-trained baseline model as warm start, the additional training cost is not high. The pre-training takes 60 GPU days on En→De; the proposed soft prototype takes extra 20 GPU days with warm start, resulting in total 1.3x time with significant performance improvement.

**Study on Parameter Reuse.** We study the influence of parameter reuse in our approach (i.e., the parameters of `Enc` and `Net`). We compare the performances on WMT2014 En→De under two settings: (A) *Shared* setting with `Enc = Net` and one $F_C$; (B) *Non-Share* setting with independent `Enc` and `Net` and two separate $F_C$'s from Eqn.(4). The BLEU scores of *Shared* and *Non-Share* settings are 29.46 and 29.45 respectively, which are almost identical. This indicates that the improvements of our approach are brought by the soft prototype instead of introducing more model parameters. We set `Net=Enc` in our experiments to minimize the total number of parameters for the rest of the experiments. However, it's worth pointing out that this is not an inherent setup of the proposed approach. `Net` can generally share or not share parameters with `Enc`, or even use a different network architecture (e.g. with different number of layers / hidden dimensions, etc).

**Case Study.** We present two examples from WMT2014 En→De task to illustrate how the method produced better translation results. The examples are presented in Table 2, including the source sentence, the reference (i.e. the ground truth translation), the translation given by the standard Transformer (*Transformer*), Transformer with hard prototype (*DelibNet*) and soft prototype (*SoftProto*). We can see from the two examples that the proposed framework with soft prototype improves the translation quality in two ways:

(A) *Better capture of the content of input sentences.* In the first example, the semantic of "how to talk with their child" is missing by the Transformer baseline, while the two methods with prototype successfully capture the meaning. Similarly in the second example, *SoftProto* is the only one that correctly reveals the relation that "Christian Ströbele is Green Party MP", whereas in the standard

bootstrap-hypothesis-difference-significance.pl

Table 2: Translation examples from WMT2014 En→De task. We use the italic fonts in red to indicate the sentence pieces that are not accurate, use the bond fonts in blue to highlight the accurate translations, and underline the corresponding content in the source sentence.

| | |
|---|---|
| Source | Parents should not hesitate to get guidance from their pediatrician about how to talk with their child . |
| Reference | Eltern sollten nicht zögern , sich von ihrem Kinderarzt beraten zu lassen , **wie sie mit ihrem Kind sprechen sollten .** |
| Transformer | Eltern sollten nicht zögern , sich von ihrem Kinderarzt beraten zu lassen . |
| DelibNet | Eltern sollten nicht zögern , sich von ihrem Kinderarzt beraten zu lassen , wie *man* mit ihrem Kind spricht . |
| SoftPrototype | Eltern sollten nicht zögern , sich von ihrem Kinderarzt beraten zu lassen , **wie sie mit ihrem Kind sprechen können .** |
| Source | According to Green Party MP . Hans @-@ Christian Ströbele , his surprising meeting with Snowden in Russia addressed the conditions under which the former Secret Service employee would make a statement to a German District Attorney or before an investigation committee . |
| Reference | Nach Darstellung des Grünen @-@ Bundestagsabgeordneten Hans @-@ Christian Strö-bele ging es bei seinem überraschenden Treffen mit Snowden in Russland darum , unter welchen Bedingungen **der Ex @-@ Geheimdienstmitarbeiter** bei einer deutschen Staat-sanwaltschaft oder vor einem Untersuchungsausschuss aussagen würde . |
| Transformer | In seinem überraschenden Treffen mit Snowden in Russland befasste sich Hans @-@ Christian Ströbele nach Ansicht des Abgeordneten der *Grünen Partei* , mit den Bedin-gungen , unter denen *ehemalige Mitarbeiter* des Geheimdienstes vor einem deutschen Bezirksstaatsanwalt oder vor einem Untersuchungsausschuss Stellung nehmen würde . |
| DelibNet | In seinem überraschenden Treffen mit Snowden in Russland erörterte Hans @-@ Christian Ströbele die Bedingungen , unter denen **der** *ehemalige Mitarbeiter* des Geheimdienstes eine Erklärung vor einem deutschen Bezirksstaatsanwalt oder vor einem Untersuchungsauss-chuss abgeben würde . |
| SoftPrototype | Laut dem **Grünen @-@ Abgeordneten** Hans @-@ Christian Ströbele ging es bei seinem überraschenden Treffen mit Snowden in Russland um die Bedingungen , unter denen *ein* **ehemaliger Geheimdienstmitarbeiter** vor einem deutschen Bezirksanwalt oder vor einem Untersuchungsausschuss eine Erklärung abgibt . |

translation, it is not clear that the person is the Green Party MP. We conjecture that with the soft prototype, the global information at the target side is enhanced, which is particularly helpful for the generation of longer and harder sentences.

(B) *More accurate selection of words*. In the first example, "sie" is a better translation than "man" from output of *DelibNet*. In the second example, while both "mitarbeiter" and "geheimdienstmitarbeiter" has the meaning of "employee", the latter (from *SoftProto*) accurately expresses the meaning for "secret service employee". However, *SoftProto* still has its limitation. For example, the word "ein" from "ein ehemaliger Geheimdienstmitarbeiter" (a former secret service employee) should be "der" (the) instead, since it refers to Snowden in particular.

**Quantitative Study w.r.t. Sentence Length.** We further conduct the following error analysis on the WMT2014 En→De task. We break down sentences in the dev set into two groups: very long sentences (length $> 40$) vs. very short sentences ($< 20$) based on the length of source sentences, and measure the performances on the two subsets. Our method achieves $0.37$ BLEU gain on the short subset, and $1.57$ BLEU gain on the long subset over the baseline, which roughly further demonstrates that our method is particularly helpful for the generation of longer and harder sentences.

## 4.2 Semi-supervised NMT

We study the semi-supervised setting with both bilingual and monolingual data. In this setting, we directly apply our framework to the state-of-the-art translation systems to verify whether our framework can generally contribute to further improving the translation quality.

Table 3: Detokenized BLEU scores on various test sets of WMT En→De translation. (S) stands for single model performance and (E) stands for ensemble.

| Model | Newstest14 | Newstest15 | Newstest16 | Newstest17 | Newstest18 |
|---|---|---|---|---|---|
| FAIR [2] (S) | $32.67 \pm 0.16$ | $33.89 \pm 0.21$ | $37.04 \pm 0.17$ | $31.86 \pm 0.22$ | $44.63 \pm 0.13$ |
| **SoftPrototype (S)** | $33.25 \pm 0.21$ | $34.67 \pm 0.26$ | $38.35 \pm 0.25$ | $32.88 \pm 0.18$ | $46.28 \pm 0.27$ |
| MS-Marian [6] (E) | − | − | **39.6** | 31.9 | **48.3** |
| FAIR [2] (E) | 33.69 | 34.73 | 37.99 | 32.80 | 46.05 |
| **SoftPrototype (E)** | **34.0** | **35.7** | 39.4 | **33.7** | 48.1 |

**Configuration** We work with 6 powerful En→De translation models released by FAIR[8] [2], which have been well trained on 128 GPUs in a semi-supervised setting, with 5.18M bitext and 226M monolingual sentences used as training data. We leverage these models to generate the probability $g^\kappa$ ($\kappa = 3$) and as initialization for our modified Transformer with soft prototype. We train our models on 8 M40 GPUs with $4.5M$ bitext from WMT2014 En→De for another $1.5$ days.

We report the single model performances with mean and standard derivation of the six models, and ensemble results of all different runs. We compare the model fine-tuned with soft prototype, with two other systems representing the state-of-the-art, including *FAIR* [2] and the WMT2018 champion system *MS-Marian* [6]. The translations are generated with beam size of 5 and length penalty 1.0, and all models are evaluated on various test sets (Newstest2014-2018) with sacreBLEU[9].

**Results** We can observe from the results in Table 3 that: (1) Our framework can further improve the model performances with a large margin. Comparing with [2], the performances of single models are boosted by 1.0 BLEU score on average, and that of ensemble models are improved by up to 2 BLEU score. (2) We achieve state-of-the-art results on Newstest2014, 2015 and 2017. These observations demonstrate that our proposed framework with target side soft prototype is capable of further enhancing the translation qualities, even for the very powerful translation systems.

## 4.3 Unsupervised NMT

We apply our framework to the scenario of unsupervised NMT, a recent and popular research direction that aims to learn the translation models without access to any parallel training data.

**Datasets** We experiment on WMT2016 En↔De translation. We use 50M monolingual sentences for each language from Newscrawl 2007-2017 as training data following [10], and report `multi-bleu` scores on Newstest2016.

**Configuration** We use the *Base* Transformer setting following [10], with the number of layers, embedding dimension, the hidden dimension and filter sizes set as 4, 512, 512 and 2048 respectively. We set $\kappa = 3$ for $g^\kappa$.

**Results** The results of NMT based unsupervised translation are shown in Table 4. The baseline [10] performances are improved by our method by large margin with over 1.5 BLEU for both directions. Meanwhile, we are aware that leveraging statistical machine translation

Table 4: Results of Unsupervised NMT on WMT2016 En↔De. † denotes our own implementation, which is slightly better than reported values in [10].

| Model | En→De | De→En |
|---|---|---|
| [9] | 9.64 | 13.33 |
| [19] | 10.86 | 14.62 |
| [10] | 17.16 / 17.64† | 21.00 / 22.24† |
| **SoftPrototype** | **19.23** | **23.78** |

(SMT) techniques could further improve unsupervised NMT [13, 10], and will leave the combination of our framework with SMT as a future work.

# 5 Conclusion and Future Work

In this work, we propose a general framework with a target side soft prototype. We discuss how the previous two-pass decoding based approaches can be adapted to our framework with different prototypes, and propose an alternative way to generate the prototype with expected semantic representations. Experiments show that our framework brings significant improvements in generation quality, with reasonable increase in storage and inference time. We evaluate our method with extensive empirical studies on multiple different tasks, and achieve state-of-the-art results on WMT2014, 2015 and 2017 En→De translation.

The proposed framework with soft prototype is general and widely applicable to almost all model architectures for sequence generation tasks not limited to NMT, and we will explore them in the future. Different approaches can be used to generate the soft prototype. Exploring how to construct the soft prototype with better quality in a context-aware manner efficiently, is another interesting and valuable future work direction.

## Footnotes

[3]For $y'$ of different sizes, we pad all the $y'$ to the largest size with padding elements as zeros.

[4]In fact, due to the high cost of searching in large sequence space, these retrieval based approaches are not applicable to tasks with tremendous number of candidates like NMT. Therefore, we focus on discussing the generation based approaches, in particular the representative model DelibNet [18], in the rest of this paper.

[5]Data filtration rules are from `https://goo.gl/SG9HLh`.

[6]`https://github.com/moses-smt/mosesdecoder/blob/master/scripts/generic/multi-bleu.perl`

[7]https://github.com/moses-smt/mosesdecoder/blob/master/scripts/analysis/

[8]`https://dl.fbaipublicfiles.com/fairseq/models/wmt18.en-de.ensemble.tar.bz2`.

[9]sacreBLEU signature: `BLEU+case.mixed+lang.en-de+numrefs.1+smooth.exp+test.wmt${SET}+tok.13a+version.1.2.11`, with SET$\in\{$14,15,16,17,18$\}$.

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
