[Supplementary Material · supplementary.pdf]

# Neural Machine Translation with Soft Prototype (Supplementary Document)

## A  More Details of the Formulation

$F_C(q, K, V)$ consists of three inputs $q, K, V$, where $q \in \mathbb{R}^d$, $K = \{k_1, k_2, \cdots, k_m\}$, $V = \{v_1, v_2, \cdots, v_m\}$, and $m$ is the size of the set $K$ and $V$. For any $i \in [m]$, $k_i, v_i \in \mathbb{R}^d$. Mathematically,

$$F_C(q, K, V) = \sum_{i=1}^{m} \alpha_i v_i; \ \alpha_i \propto \exp(W_q q + W_k k_i), \tag{1}$$

where $\alpha_i \geq 0$, $\sum_{i=1}^{m} \alpha_i = 1$, and $W$'s are the parameters to be learned. $F_S$ is the same as $F_C$, with different parameters $W.$'s.

For any $x \in \mathbb{R}^d$,

$$F_N(x) = \max(xW_1 + b_1, 0)W_2 + b_2, \tag{2}$$

where $\max$ is an element-wise operator, $W$'s and $b$'s are the parameters to be learned.

## B  Empirical Analysis of Soft Prototype

### B.1  Study on Parameter Reuse.

We study the influence of parameter reuse in our approach (i.e., the parameters of `Enc` and `Net`). On WMT2014 En→De, we compare the performances under two settings:

1. *Shared* setting with `Enc = Net` and one $F_C$;

2. *Non-Share* setting with independent `Enc` and `Net`, and two separate $F_C$'s from Eqn.(4).

The BLEU scores of *Shared* and *Non-Share* settings are $29.46$ and $29.45$ respectively, which are almost identical. This indicates that the improvements of our approach are brought by the idea of soft prototype, instead of introducing more model parameters.

### B.2  Study on Values of $\kappa$.

We study the performances of our approach with respect to different values of $\kappa$. We build probabilistic generator $g^\kappa$ with $\kappa = \{1, 2, 3, 5, 10, 20\}$ on the WMT2014 En→De dataset, and train the models with different prototype.

As we can see from the results in Figure 1, the soft prototype works the best with relative small values for $\kappa$ (e.g. $\kappa \leq 10$), while the larger value like $\kappa = 20$ hampers the model performance. Our conjecture is that large values for $\kappa$ encourage better information converge than small values, yet bring in more noise into the model at the same time, which is deleterious for the model training. With $\kappa$ set in a reasonable range, the model can benefit from adequate additional information with a tolerable level of noise. We use $\kappa = 3$ in the rest of our experiments to best utilize the proposed approach with a minimal increase in storage and inference cost.

Figure 1: BLEU scores of WMT2014 En→De with different $\kappa$.

## B.3 Study on Re-initializing Probability Generator $g$

After obtaining a better model, we reinitialize a generator $g^\kappa$ with our best model on En→De (i.e., BLEU= 29.46), and achieve 0.14 BLEU improvement compared to the previous best model (i.e., BLEU= 29.60). This shows that a better $g$ is helpful to improve performances.