[Reviews · NeurIPS 2019]

Reviewer 1



The paper addresses the problem of lacking global context on the target side when generating output from left to right. It starts by pointing out that other methods which do decoding in more than one step (by e.g. first generating the sequence right to left) are inefficient computationally. The work proposes a soft prototype, which combines multiple potential outputs on the target side and is more efficient than previous work. The motivation is very clear and the paper is, for the most part, well written and explained. The approach is not ground-breaking, however it is a good approach that makes sense for addressing the problem considered and if more efficient than previous work. Better MT quality improvements would have made a stronger case, however the method is clearly better than previous work and the improvements are stable across different settings. Specific questions/comments: - Section 3.2: how does this become non-autoregressive? By not modeling any dependency between the rows of G_y (i.e. between gx(i))? I did not find the discussion at the end of 3.1 to be very helpful to understanding 3.2 - 3.2 seems the simpler formulation. - Table 1: Does the number of params/inference time refer to the French or German model? The different vocabulary sizes have to lead to different number of parameters. - Minor: Not sure I fully understand the choice of term “prototype” in this work - the term makes more sense when the data is retrieved (citations 4 and 5 in the paper).

Reviewer 2



- The paper is based on the similar motivation of previous studies on NMT with prototypes. The actual method, that uses extra encoders for additional information, looks also a usual way of the multi-source NMT settings. Though the originality is limited, the whole construction of the model looks well. In the formulation of the method, they discusses the difference between the proposed method and existing studies. - No enough analysis of the results found. The "Case Study" section does not distinguish their improvement against random changes. - The method is enough simple to reproduce in any existing toolkits, and readers may be able to implement the same method by reading the paper (though there are some formulation mistakes). - The translation quality improvement looks quite good, though they tested the methods by only syntactically similar language pairs.

Reviewer 3



Neural Machine Translation with Soft Prototype The paper suggests to equip a neural machine translation system with a soft prototype in order to provide global information when generating the target sequence. The suggested approach shares similarities with a multi-pass decoding strategy such as in deliberation networks, however, with the difference that the prototype is not a hard sequence of tokens but a soft representation. To achieve fast inference speed and only a small increase in terms of model parameters compared to the baseline system, the authors share the parameters between the Encoder network and the additional network used to encode the soft prototype. Experiments are conducted for three different setups on the WMT EnDe and EnFr tasks: a supervised, a semi-supervised and an unsupervised setting. The proposed technique yields gains between 0.3 and 1.0 BLEU points depending on the setup over their corresponding baselines and are claimed to achieve new state-of-the-art results. Overall, this is a good paper with nice results given the fairly small increase of model parameters, and I would like to see this paper at the conference. I think that the exposition of the soft prototype could be improved by putting more emphasis onto the actual model that the authors finally choose to use. The initial formulation is quite general and fairly different compared to what is eventually used in the paper, and it would be interesting to see results on different network incarnations of the soft prototype framework. E.g., one natural question is how the network would fair if the network $Net$ would not share its parameters with the encoder. Besides that I have only a few suggestion, comments and questions as stated below. The claim that the suggested approach yields state-of-the-art results is not entirely true, as there have been other techniques that surpass the best BLEU score results reported in this paper. E.g., the publication "PAY LESS ATTENTION WITH LIGHTWEIGHT AND DYNAMIC CONVOLUTIONS" by F. Wu et al. reports a BLEU score of 29.7 on the EnDe WMT translation task. L260: The analysis of the first German sentence is not entirely correct: "sie" has no corresponding word on the source side. The phrase "how to" is the one that corresponds to "wie sie". A more literal source sentence for the generated German target would have been "how [they should] talk". The word "their" corresponds to "ihrem". Editorial comments: L275: replace "derivation" with "derivative" ------- Thanks to the authors for sharing their responses. Since I have considered this submission to be among the top 50% of accepted submissions, I still stand with my original rating: This paper should get accepted for publication at NeurIPS 2019.

[Author Response · NeurIPS 2019]

Thank all the reviewers for their valuable comments and suggestions!

**Response to Reviewer #1**

1. Section 3.2: Each row of $G_y = g(x_i)$ is a distribution independent from other rows, and therefore is in a non-autoregressive manner (see Line 126-129 for more details). At the end of Section 3.1, we show that previous works [9,12,22,5,4] implement $G_y$ as a matrix with one-hot rows due to the complexity brought by the autoregressive computation. This motivates us introducing non-autoregressive generation for the prototype in Section 3.2.

2. Table1: The number of parameters and inference time refers to the WMT14 En→De translation model.

3. Prototype: In [4,5], a prototype refers to an intermediate target sequence in the that will be further refined. We follow the usage of term "prototype" with similar motivation. Different from the retrieved/generated sentences that can be regarded as "hard" prototypes, we introduce the prototype in a soft manner where the expectation of the intermediate sequence is calculated. Furthermore, the term "prototype" also refers to the "mean" (i.e., average) of a set of points in a cluster [*]. In our paper, it refers to the average of embeddings, matching the sense of "prototype" used in clustering. [*] Tan, Pang-Ning. Introduction to data mining. Pearson Education India, 2018.

**Response to Reviewer #2**

1. Eqn.(3) is an expectation of embeddings that carries the first-order statistical information of all potential translations. The mean of vectors geometrically represents the centroid of multiple vectors, so it is a meaningful representation.

2. $g^\kappa$ is normalized by keeping only top-$\kappa$ largest probabilities and then scaling. For example, for $g = (g_1, g_2, g_3)$ with $\kappa = 2$ and $g_1 > g_2 > g_3$, $g^\kappa = (g_1/(g_1 + g_2), g_2/(g_1 + g_2))$.

3. Parameter reuse: (a) We set `Net` = `Enc` in our experiments only to minimize the total number of parameters. Please note that it is not *inherent* setup of the proposed method (Line 168-169). `Net` can generally share or not share parameters with `Enc`, or even use a different network architecture (e.g. with different number of layers / hidden dimension, etc). We also show in Appendix B.1 that for WMT En→De, the proposed method achieves comparable performance with/without parameter reuse. (b) We work on IWSLT2014 English→Chinese, a distant language pair following your suggestion. We use the Transformer with a 6-layer encoder and decoder, with the hidden dimensions and filter sizes set as $512$ and $1024$ respectively. The baseline is $15.4$ BLEU score and the proposed method achieves $15.8/16.0$ BLEU with/witout parameter reuse. We would like to highlight that one advantage of the proposed approach is that it's general, and in the future we will further evaluate it with more syntactically different language pairs.

4. Token-level translation: We use token-level mapping for the maximum efficiency. We agree that it makes less sense for the "middle" tokens. However, the impact would be relatively small given that: (a) the vocabulary and training corpus is dominant by the standard words rather than the "middle" tokens. For example, for WMT14 En→De, over $65\%$ vocabulary are standard words and they make up for over the $88\%$ of total word frequency in the training corpus; (b) The soft prototype $R$ is fed to `Net` and encoded into higher-level contextual representations, which can intuitively provide rich global information that helps the decoder decision making.

5. Case study: (a) Our goal of the case study is not to claim that the proposed method is always beneficial in the two ways we described, but to use the two examples to illustrate how exactly the method produced better translation results in those two randomly picked examples. For this purpose, our analysis is useful in that it revealed two benefits in the two examples. (b) We agree that it is better to more systematically analyze the benefit of the proposed method. However, manual examination of a very large number of examples in the same way as we did for the two examples in case study is infeasible. So we have done the following error analysis: we break down the sentences into two groups: very long sentences (length $> 40$) vs. very short sentences ($< 20$) based on the length of source sentences, and measure the performances on the two subsets. Our method achieves $0.37$ BLEU gain on the short subset, and $1.57$ BLEU gain on the long subset over the baseline, which roughly shows that our method is "particularly helpful for the generation of longer and harder sentences". We will add the systematic analysis of benefits and errors as a future work.

**Response to Reviewer #3**

1. Thanks for your suggestions. We will revise the writing in the next version. As for the different network incarnations, we studied the parameter reuse and found it achieves comparable performance on En→De translation with/without parameter sharing (Appendix B.1). We also tried a shallower network for `Net` with a 2-layer Transformer encoder, and achieve 29.29 BLEU in En→De translation. We will explore more on different network incarnations in future work.

2. We achieve the state-of-the-art results in Newstest 2014, 2015 and 2017 in the semi-supervised setting in Section 4.2 (detokenized sacreBLEU reported in Table 3), which are the best performances so far under the same training data setting to the best of our knowledge.

3. L260: Thanks for the detailed suggestion. We will revise the analysis and make it more accurate.

[Meta-Review · NeurIPS 2019]

Good paper. Accept.